# Physiological State of Therapy Dogs during Animal-Assisted Activities in an Outpatient Setting

**DOI:** 10.3390/ani10050819

**Published:** 2020-05-09

**Authors:** Stephanie D. Clark, François Martin, Ragen T.S. McGowan, Jessica M. Smidt, Rachel Anderson, Lei Wang, Tricia Turpin, Natalie Langenfeld-McCoy, Brent A. Bauer, Arya B. Mohabbat

**Affiliations:** 1Section of Integrative Medicine and Health, Division of General Internal Medicine, Mayo Clinic, Rochester, MN 55901, USA; sdclark4@icloud.com (S.D.C.); smidt.jessica@mayo.edu (J.M.S.); bauer.brent@mayo.edu (B.A.B.); 2Nestlé Purina Research, St. Louis, MO 63102, USA; francois.martin@rd.nestle.com (F.M.); ragen.trudelle-schwarzmcgowan@rd.nestle.com (R.T.S.M.); rachel.anderson@rd.nestle.com (R.A.); lei.wang@rd.nestle.com (L.W.); patricia.turpin@rd.nestle.com (T.T.); natalie.langenfeld-mccoy@rd.nestle.com (N.L.-M.)

**Keywords:** animal-assisted activity, therapy dogs, emotional state, wellbeing, physiology

## Abstract

**Simple Summary:**

Therapy dogs and their benefits to human health have been studied extensively, but investigating the animal’s welfare during therapy sessions is limited. Moreover, existing literature has mixed results as to the emotional state and wellbeing of the therapy animals. This study focuses on 19 therapy dogs’ wellbeing during animal-assisted activities, by evaluating their heart rate and heart rate variability, salivary cortisol and oxytocin, and ear temperatures. The results demonstrated that the dogs’ wellbeing was not negatively affected during these visits. Moreover, pre- and post-visit physiological indicators remained stable and some of the dogs’ parameters suggested the therapy dogs may have been in a more relaxed state at the end of the session.

**Abstract:**

Therapy dogs are increasingly being incorporated into numerous clinical settings. However, there are only a handful of studies that have focused on the impact of animal-assisted activity or therapy sessions on the wellbeing of the therapy dogs. Furthermore, these studies show mixed results. The goal of this study was to provide an in-depth picture of the effects of these interactions on the dogs involved by considering multiple physiological measures known to be associated with emotional state (continuous heart rate, heart rate variability, pre- and post-session tympanic membrane temperatures, and salivary cortisol and oxytocin concentrations). Nineteen Mayo Clinic Caring Canine therapy dogs completed five 20-minute animal-assisted activity (AAA) visits each in an outpatient clinical setting (Mayo Clinic Fibromyalgia and Chronic Fatigue Clinic). From a physiological perspective, the dogs showed a neutral to positive response to the AAA sessions. Heart rate (HR) was significantly lower at the end of the session compared with the beginning of the session (F = 17.26, df1 = 1, df2 = 29.7, *p* = 0.0003). The right tympanic membrane temperature was lower post-session (F = 8.87, df1 = 1, df2 = 107, *p* = 0.003). All other emotional indicators remained stable between pre- and post-session. These results suggest that the dogs involved were not negatively affected by their participation in the AAA. Moreover, there was some evidence suggesting the dogs may have been in a more relaxed state at the end of the session (lower HR and lower right tympanic membrane temperature) compared to the beginning of the session.

## 1. Introduction

The focus on animal wellbeing has increased significantly over the last several decades. This includes the wellbeing of animals involved in activities designed to improve human physical and psychological health such as animal-assisted therapy (AAT) (“Goal-oriented, planned and structured therapeutic intervention directed and/or delivered by health, education or human service professionals”, IAHAIO White Paper 2018, [1] p. 5) and animal-assisted activity (AAA) (“Planned and goal-oriented informal interaction and visitation conducted by the human-animal team for motivational, educational and recreational purposes”, IAHAIO White Paper 2018, p. 5). However, wellbeing standards for dogs used in AAA may be particularly difficult to establish. What constitutes an acceptable level of wellbeing is largely defined by societal values, and these values are changing and evolving as society changes. In many current societies, dogs no longer have a utilitarian status; rather, dogs are considered part of the human family and their wellbeing standards reflect this new status.

Groups involved in research, education, and the practice of AAA and AAT, such as the International Association of Human-Animal Interaction Organizations (IAHAIO), Pet Partners, and the American Veterinary Medical Association (AVMA) have published general guidelines and best practices to help ensure the wellbeing of therapy animals. IAHAIO’s White Paper (2018) entitled “Definitions for animal assisted intervention and guidelines for wellness of animals involved” states that the animals should be “in good health, both physically and emotionally and that (they) enjoy this type of activity” (p. 7). In their position statement on the health and welfare of therapy animals, Pet Partners posits that the handler is the animal’s best advocate and that “being an animal’s advocate requires making decisions based on the preference of the animal” [2] (p. 1). The AVMA guideline on animal-assisted interventions (AAI) (2020) [3] stipulates that the person responsible for the intervention, “should be educated about behavioral signs that might indicate that an animal is not enjoying an activity associated with AAI.” Taken together, these statements point to the fact that not only is it expected that therapy dogs should not suffer from their participation in AAA, but that they should also be in a state of positive wellbeing, while involved in these activities. Work by animal welfare and behavior scientists in the field of AAA over the last decade or so has helped objectively measure the wellbeing of these animals. The present work aims at contributing to the body of knowledge in this field of study.

Therapy dogs are asked to perform various tasks, such as interacting with unfamiliar people in unfamiliar environments [4,5], and there is the potential for these interactions to cause stress for therapy animals [6]. For example, during AAA sessions dogs are exposed to novel items such as wheelchairs, crutches, sudden noises, and unusual substrates (e.g., tile, stairs, and iron grid) [7,8]. Furthermore, inadequate environmental conditions, high temperatures, lack of space, and age of clients have been reported as potential factors negatively influencing the wellbeing of therapy dogs [9]. As the utilization and versatility of AAA expands, there is a call for investigating the direct effects of these activities on the wellbeing of the therapy animals. However, there are relatively few studies focused on the potential impact of AAA sessions on the wellbeing of therapy dogs and, collectively, these studies provide somewhat mixed results [8].

Measuring stress in animals is complex [10] and understanding the effects of AAA on the wellbeing of therapy dogs is challenging. A review of nine papers related to AAA studies reported a large variation in terms of study designs (case reports, original research, and Doctoral and Master’s theses), number of participants and dogs, disabilities, type of activities performed, and duration and frequency of visits [8]. The wellbeing indicators used between studies also varies, including salivary, fecal, urine, or hair cortisol; heart rate (HR); questionnaires; interviews; behavior observation; and cognitive tests. Given the wide array of contexts and methodologies, some of these studies reported changes usually associated with negative emotional states (e.g., increased cortisol levels, stress-related behaviors), while others found no significant changes in emotional indicators, or reported changes usually associated with positive emotional states [8].

Clark et al. [11] studied the impact of hospital visits on therapy dogs pre- and post-interaction and at different visit frequencies. Dog salivary cortisol was compared post-interaction to baseline for each different visit frequency: twice a week, once a week, twice over a four-week period, and once over a four-week period. Dogs had significantly lower post-interaction salivary cortisol concentrations when they performed AAA twice a week compared to baseline concentrations. In addition, while not statistically significant, dog cortisol concentrations were lower post-interaction compared to baseline when dogs visited twice as well as once within a four-week period. These results suggest that therapy dogs were not negatively affected by AAA and, in fact, appeared to be less aroused and calmer when they participated in more frequent AAA sessions.

McCullough et al. [12] measured cortisol and behavior in therapy dogs participating in AAA with children with cancer. Twenty-six handler–dog teams participated in the trial. Over four months, the teams were paired with children recently diagnosed with cancer for weekly AAA sessions lasting about 20 min. Salivary cortisol was measured at several time points. The dogs’ behavior during the session was recorded and later coded for affiliative behaviors, moderate stress behaviors, and high stress behaviors. The authors found minimal signs of distress during sessions. There was no difference in cortisol concentrations between baseline and 20 min after the start of AAA. Cortisol concentrations did not increase over time, suggesting that the dogs were not negatively affected by their participation in AAA.

Melco et al. [13] studied the impact of AAT on the wellbeing of nine therapy dogs. The dogs and their owners participated in sessions designed for children with attention deficit hyperactivity disorder (ADHD) that included activities such as social skills training, dog training, and reading and writing in the company of the therapy dogs. Twice a week, for three weeks, the dogs participated individually in small group (3–4 children) AAT sessions divided into five 20-min activities: (1) calming activity; (2)–(4) therapy exercises; (5) calming activity. Each dog participated in six sessions. After each activity, the dogs’ HR was auscultated and a saliva sample was taken for cortisol analysis. The dogs’ behavior was also observed during the five activities. Using a one-zero interval sampling, observers recorded behavior usually associated with stress (e.g., lip licking and panting) every 20 sec for ten minutes for each of the five activities. Very few stress-related behaviors were recorded during the activities and there was no change in HR or in salivary cortisol concentration between the activities or over the 3-week period. The researchers concluded that the dogs involved in these AAT activities experienced minimal stress.

King et al. [14] enrolled twenty-seven certified therapy dogs in a study designed to measure the impact of a short quiet time (e.g., toy chewing, petting, and talking to the dog) midway through a two-hour AAT “shift” on the wellbeing of the dogs. Each team participated in one experimental session (quiet time) and one control session (no quiet time). Salivary cortisol concentrations were measured (1) in the morning at home on a non-AAT day, (2) at the end of the first AAT hour, and (3) 30 min after the end of the AAT session. After the AAT sessions, the dogs were brought to the AAT office (a room they were familiar with) and observed for one minute for signs of stress (e.g., pacing and panting). The quiet time did not have an effect on salivary cortisol levels or the stress-related behaviors. However, the younger dogs and the dogs with less experience in AAT tended to show more stress-related behaviors.

Palestrini et al. [15] measured the behavior and HR of a trained and experienced 7-year-old female Golden Retriever while she was interacting individually with 20 children approximately two hours after they received surgery. The dog’s HR remained within normal range during the 20-min AAT sessions. Panting was observed 28.35% of the time, lip licking for 5.65% of the time, and yawning 1.25% of the time. The authors concluded that it was difficult to determine if the excessive panting was due to the ambient temperature, in response to stress, or to positive arousal. In addition, lip licking, yawning, grooming, or avoidance behavior did not correlate with AAT sessions with significant child interaction or with sessions where the child did not interact with the dog.

Haubenhofer and Kirchengast [16] compared the salivary cortisol concentrations of 18 therapy dogs when they were at home versus when they were performing AAA/AAT session. They found that the cortisol concentrations were higher on AAA/AAT days and that dogs that completed more sessions had higher cortisol concentrations. They also found that dogs that were involved in shorter sessions (one hour) had higher cortisol concentrations compared to dogs that were involved in longer sessions (up to eight hours). The authors explained this result by the fact that the shorter sessions were characterized by the dog handlers as more intense and more demanding for the dogs than the longer sessions (with breaks and quiet times). The time of day also had an impact on cortisol concentrations. When the sessions were conducted before 2 PM, dogs had higher cortisol concentrations post sessions; when the sessions were conducted after 2 PM, dogs had lower cortisol concentrations post sessions. The authors concluded that the AAA sessions were physiologically arousing for the dogs, but could not attribute a valance to the dogs’ emotional experience.

De Carvalho et al. [17] compared salivary cortisol, HR, and respiratory rate (RR) of 19 experienced therapy dogs at home versus right after AAI sessions. Salivary cortisol concentration values were higher after AAI sessions. However, because of the experimental design, it was not possible for the authors to know if this meant that the dogs were more stressed or more aroused. Time of day, number of sessions per week or their duration, number of beneficiaries present, sex or breed of the dogs, and being on or off leash did not have an effect on cortisol concentrations. HR and RR values were also higher post AAI sessions, compared to at home. However, most home and post-session values remained between minimum and maximum normal values. The type of AAI sessions had an effect, with dogs participating in AAT sessions having lower HR compared to dogs participating in AAA sessions. The duration of the transportation from home to the AAA site also had an effect. Dogs that traveled for more than 50 min had higher HR, compared to dogs with shorter transportation durations. Some of the dogs in this study had values (either at home, or after AAI sessions) that were outside of normal limits. The authors explained these findings by external variables (e.g., high ambient temperatures, a dog being afraid of car rides, and the presence of multiple children surrounding a therapy dog). De Carvalho et al. [17] concluded that the welfare of the therapy dogs was not significantly affected by the AAI sessions, but they pointed out the importance of performing AAI in appropriate conditions.

Based on the literature reviewed, it appears that several factors may potentially influence the wellbeing of therapy dogs, such as frequency and/or duration of sessions, characteristic of participants, or the training level and experience of the therapy dogs. Sometimes, the results were counterintuitive (e.g., in some instances, more frequent visits or longer duration of visits were associated with positive wellbeing). In addition, in some cases, it was difficult for the authors to interpret their results because too few indicators of wellbeing were used. However, the literature also suggests that it is reasonable to expect no major negative impact when AAA sessions are conducted with well-trained therapy dogs in a controlled setting.

In this study, we evaluated the emotional state of therapy dogs in response to AAA in an outpatient, clinical setting (Mayo Clinic Fibromyalgia and Chronic Fatigue Clinic). We measured changes in various physiological parameters, using noninvasive methodologies: salivary oxytocin, salivary cortisol, tympanic membrane temperatures, and cardiac activity. To our knowledge, this combination of methodologies (including a new salivary oxytocin analysis developed for this project) has not been used before. As the canines used in this study were well-trained, registered therapy dogs, we predicted that the AAA sessions would have no negative effect on their emotional state (i.e., the physiological markers’ values would remain unchanged between pre- and post-session).

## 2. Materials and Methods 

### 2.1. Study Design

This study was part of a larger effort aimed at assessing the impact of AAA on people with fibromyalgia (FM) and therapy dogs [18]; however, the focus of this paper will be solely on the therapy dog portion. In the larger study, two-hundred-and-twenty-two (222) human subjects between the ages of 18 to 76 years were randomly assigned to either the treatment group (n = 111) (a 20-min session with a therapy dog and volunteer) or the control group (n = 111) (a 20-min session with just the volunteer). The human participants were patients attending Mayo Clinic’s Fibromyalgia Treatment Program, a 1.5-day, multidisciplinary, outpatient treatment program, staffed by physicians from the Division of General Internal Medicine. The program consists of individual and group sessions, spanning topics such as diagnostics, pathophysiology, and medication and non-medication treatment strategies for FM. The primary aims of the treatment program are to improve physical and mental health functioning, provide evidence-based treatment options, and create a lasting treatment regimen for patients with FM. Mayo Clinic’s Institutional Review Board committee approved this study (protocol number 16-006296).

### 2.2. Therapy Dogs

This study utilized 19 therapy dogs that were from Mayo Clinic’s Caring Canine Program. The dogs were registered as therapy dogs with Alliance of Therapy Dogs, Pet Partners, Therapy Dog International, or Helping Paws. They were up to date on vaccines, deemed healthy by their veterinarian, at least one year of age, and were not fed a raw diet [19,20]. Of the therapy dogs, 13 were female and six were male. All dogs were spayed or neutered and represented various breeds (Table 1). The dogs were always accompanied by their owners (referred to as “volunteers” later in the text) and each dog completed five 20-min sessions. Mayo Clinic’s Institutional Animal Care and Use Committee (IACUC) approved this study (protocol number A00002194-16).

Approximately 15 min after arriving at the hospital and before meeting the patients with FM in the exam room, the dog and volunteer were brought to an adjoining room where the dog’s saliva was collected, bilateral tympanic membrane temperatures were taken simultaneously, and a heart rate monitor was placed on the dog by a study team member. At the end of the 20-min session, the dog and volunteer left the exam room and went back to the adjoining room. Post-session saliva and tympanic membrane temperatures were collected from the dog and the heart rate monitor was removed.

### 2.3. Volunteers

A total of 19 volunteers participated in the study: 15 women and four men. All of the therapy dog teams volunteered at Mayo Clinic’s hospital on a regular basis and were included in this study on a voluntary basis. The volunteers were asked to conduct themselves in a professional manner and to uphold the Mayo Clinic Volunteer and Caring Canine program rules, including patient privacy, dressing professionally, and ensuring that their dog was properly groomed. The volunteers were instructed to talk only about neutral topics of conversation. A conversation starter list was provided to the volunteers at the beginning of the study; topics of conversation included, weather, travel, hobbies, movies, and books. Conversations about the patient’s FM or other medical conditions were not allowed. All volunteers were greater than 18 years of age (mean = 53.1; min = 27; max = 74) and had been volunteering with their therapy dog for an average of 1.86 years (mean = 6.7; min = 1.5; max = 11). The volunteers were asked to complete a total of ten visits: five with their dogs and five by themselves.

### 2.4. Animal-Assisted Activity

Data collection occurred during the actual AAA sessions conducted with patients with FM. Therefore, the decision was made not to videotape the sessions so as not to interfere with the dynamics between the patients and the AAA team, or to cause possible embarrassment to the patients [21,22]. The sessions followed the standard Mayo Clinic Caring Canine protocol: (1) the visits were not structured in the sense that the patient could choose to interact or not to interact with the dog, (2) the dog was also free to choose whether or not it wanted to interact with the patient, and (3) the dog could freely move around the room (Table 2). 

The study interactions took place in an exam room (8’× 12’; average room temperature: 69 °F) in Mayo Clinic’s Fibromyalgia and Chronic Fatigue Clinic, which included a small exam table, four chairs, and a desk. The volunteer, the therapy dog, the patient with FM, and a study staff member were the only individuals in the room. The room was unfamiliar to the therapy dogs on their first visit. The study staff member sat at a desk in the room to monitor the interaction and was only allowed to interact with the dog if the heart rate monitor was not functioning properly (the staff member intervened 4 times to address lost signal on the heart rate monitor, and once to stop a conversation that fell outside of the approved topics). The volunteers sat in a chair or on the floor, according to their preference and were able to converse with the patient.

### 2.5. Measures and Analyses

#### 2.5.1. Salivary Cortisol and Oxytocin Concentrations

Saliva was collected from the therapy dogs with a VERSISAL^®^ (Oasis Diagnostic LLC, Vancouver, WA, USA) swab over the course of two minutes. Saliva was extracted from the collection device by centrifugation (Ultra 8D, LW Scientific, Lawrenceville, GA, USA) at 3300 rpm for ten minutes. The samples were then aliquoted into separate tubes for oxytocin and cortisol analysis. Aliquots were stored in an −80 °C freezer until overnight shipped on dry ice to Nestlé Purina’s laboratory for analysis of cortisol and oxytocin.

Salivary cortisol was analyzed on a cobas e411 (Roche Diagnostics, Indianapolis, IN, USA). The electrochemiluminescence immunoassay—Elecsys Cortisol assay—uses a competitive test principle, in which a polyclonal antibody is specifically directed against cortisol. Untreated saliva samples are used following centrifugation. Prior to the start of the analysis, an in-house assay verification was completed to evaluate limit of quantification and dilution linearity. It showed that the lower limit of quantification for the laboratory was 0.130 ug/dL, and dilution of saliva was linear, 106% linearity, up to the limit of quantification. Any value below 0.130 ug/dL was recorded as <0.130 ug/dL. Samples that did not have enough volume for analysis were diluted with Roche Elecsys Diluent Universal. For those samples that required dilution, the results took into account the dilution factor.

Oxytocin (OT) in saliva was measured by a liquid chromatography–mass spectrometry (>LC–MS) platform containing a Nexera X2 ultra-high-performance liquid chromatography (UHPLC) (Shimadzu, Columbia, MD, USA) and an AB Sciex 6500+ quadruple ion trap mass spectrometer (QTRAPMS) (AB Sciex, Framingham, MA, USA). Briefly, a 300 µL of calibration standard or saliva sample was mixed with 1.2 mL 80% aqueous ACN containing oxytocin internal standard (OT-d_5_, 1 nmol/L) in a 2 mL Eppendorf tube. The mixture was vortexed for 30 sec and then centrifuged at 15,000× *g* under 4 °C for 10 min. After the centrifugation, the supernatant was transferred into another 2 mL Eppendorf tube and dried using a miVac sample concentrator (SP Scientific, Stone Ridge, NY, USA). After completely dried, the sample was reconstituted with 50 µL 50% aqueous ACN. After another centrifugation at 15,000× *g* under 4 °C for 2 min, the supernatant was transferred to an HPLC vial. A 10 μL aliquot of the prepared sample was injected into the LC–MS for analysis. Quantification was performed by multiple reaction monitoring (MRM) of the protonated precursor molecular ions [M+H]^+^ and the related product ions. Chromatograms and mass spectral data were acquired and processed using Analyst^®^ 1.6.3 software (AB Sciex, Framingham, MA, USA [23].

#### 2.5.2. Tympanic Membrane Temperature

Tympanic ear thermometers (Braun ThermoScan^®^ PRO 6000 ear thermometer, Welch Allyn, Skaneateles Falls, NY, USA) were used to assess the temperature of both the left and right tympanic membranes, simultaneously.

#### 2.5.3. Cardiac Activity 

Cardiac activity was monitored using a Polar V800 device (Polar Electro Öy, Kempele, Findland), which includes a receiver (watch) and a transmitter (soft elastic belt with electrodes imbedded in two sections). A water-based electrode lubricant was used to enhance conductivity. Continuous cardiac monitoring was maintained throughout each session. The measured cardiac parameters included, heart rate (HR), high-frequency power (HF), low-frequency/high-frequency power ratio (LF/HF ratio), the percent of heart beats where differences between an RR interval and the previous RR interval is greater than 50 ms (PNN50), and the root square mean of the successive differences of RR intervals (RMSSD). 

The data were downloaded from the receiver to a computer using the Polar Flow application. The data were then exported from the Polar Flow (Polar Electro Öy, Kempele, Finland) application for analysis. The collected data were analyzed in two-minute intervals via Kubios HRV Standard Version 3.1.0 (Kubios Öy, Kupio, Finland). The cardiac parameters were analyzed at the beginning of the session (minutes 3 and 4) and the end of the session (minutes 17 and 18); these timeframes were selected to provide the cleanest two-minute intervals for analysis, as this gave the dog time to settle at the beginning of the session and standardized a time point before the end of the session. Cardiac data that had an artifact correction factor of greater than 10% were not included in the analysis; this was done to minimize corruption in the data, accounting for motion artifacts and interference artifacts, such as the dog’s fur.

### 2.6. Statistical Analysis

A nested linear mixed model was used to analyze differences in pre- and post-session collected values using the lme4 package in R [24]. The outcomes considered were left and right tympanic membrane temperatures, salivary oxytocin, salivary cortisol, HR, HF, RMSSD, PNN50, and LF/HF ratio. Time and volunteer sex were included in the model as fixed effect; time was the variable of interest and volunteer sex was controlled as previous studies [11,25,26,27,28] have found that the sex of the volunteers can have an effect on the dogs’ physiological and behavioral responses. A random effect for unique session was nested within therapy dog to account for the correlation between pre- and post-session values, as well as repeated measurements on each dog. Type III Sums of Squares were used to test the overall importance of a variable. Residuals were used to check model assumptions, and raw means and standard deviations are reported.

The linear mixed model and descriptive summaries were run on the raw data with statistical outliers removed, which accounted for experimental or equipment errors. Statistical outliers were identified and removed according to the 1.5 Interquartile Range (IQR) criteria, where values higher than Q3 + 1.5 × IQR or below Q1 − 1.5 × IQR were flagged and removed. Statistical significance was set at *p* < 0.05.

## 3. Results

### 3.1. Salivary Cortisol Concentrations

There were no statistically significant differences in pre- and post-session values for salivary cortisol (F = 0.08, df1 = 1, df2 = 41.6, *p* = 0.78) (Table 3).

The drop in sample size was mainly due to the limited amount of saliva that the study members were able to obtain from the therapy dogs within the two-minute window of collection before and after the session. For samples that did not have enough volume aliquoted, oxytocin assays were run first and if there was any remaining saliva it was diluted to run cortisol. 

### 3.2. Salivary Oxytocin Concentrations

There were no statistically significant differences in pre- and post-session values for salivary oxytocin in therapy dogs (F = 0.04, df1 = 1, df2 = 72.5, *p* = 0.85). 

### 3.3. Tympanic Membrane Temperature

The average right tympanic membrane temperature (°C) was significantly lower post-session than pre-session (F = 8.87, df1 = 1, df2 = 107, *p* = 0.0036). The average left tympanic membrane temperature was not significantly different post-session (F = 1.52, df1 = 1, df2 = 107.9, *p* = 0.22). 

### 3.4. Cardiac Activity

On average, HR (beats per minute) was significantly lower at the end of the session than it was at the beginning of the session (F = 7.26, df1 = 1, df2 = 29.7, *p* = 0.0003). There were no statistically significant differences between the beginning and end session values for HF, RMSSD, PNN50, or LF/HF ratio (F = 0.34, 2.45, 0.43, 0.24, and *p* = 0.56, 0.13, 0.51, 0.63, respectively).

## 4. Discussion

The present study aimed to answer the following question: Is the physiological state of well-trained therapy dogs impacted by a 20-min animal-assisted activity? We investigated this question by evaluating multiple, noninvasive biomarkers to assess the physiological response of these dogs during AAA in an outpatient clinical setting. Based on our results, we concluded that AAA did not have any negative effects on the emotional state of therapy dogs, with most physiological parameters remaining stable from beginning to the end of the session. Furthermore, HR was significantly lower at the end of the session and right tympanic membrane temperature was significantly lower post-session, suggesting that the therapy dogs may have been in a more relaxed or calm state after the session when compared with the beginning of the session.

One limitation of this study is the fact that, because we did not want to interfere with the dynamics between the patients and the AAA team, the dogs’ behavior was not recorded or scored during the AAA sessions. Behavioral observations would have provided additional information on the extent and nature of the interactions between the dogs and the patients with FM. This information would have helped better interpret the physiological data.

The mean cortisol concentration levels pre-session for the dogs in our study were comparable to concentration levels reported in the literature [29]; less is known about oxytocin in dogs. The available data show greater variation [30,31,32]. This may be due to the fact that individual dog’s oxytocin levels have a wide range. In addition, researchers are using different detection and quantification methodologies, and it is known that some methods are more sensitive than others [23]. Overall, our cortisol and oxytocin results suggest that pre-experiment variables, such as anticipation or transportation, did not affect the concentration of these hormones in the dogs.

Cortisol concentrations are modulated by the hypothalamic pituitary adrenal (HPA) axis and are a physiological indicator of a dog’s arousal level [27,33,34,35]. Because of its noninvasive nature and ease of use, salivary cortisol is widely used in dog welfare research [29]. In our study, we compared the salivary cortisol concentration of the dogs prior to and after the AAA session with a patient with FM. There was no significant difference in pre- and post-session salivary cortisol concentrations, indicating the 20-min interaction did not have a significant impact on the dogs’ arousal response. These findings are similar to those described in other previous studies [13,36,37,38,39].

Oxytocin has been shown to increase following affiliative behavior between people and dogs [30,31]. It has been utilized to assess social interactions, anxiety, and maternal behaviors [40]. Due to its involvement in stress regulation, changes in oxytocin concentrations can be utilized as a surrogate biomarker during AAA to evaluate the wellbeing of therapy dogs [41]. In our study, there was no significant difference in pre- and post-session salivary oxytocin concentrations, suggesting that the emotional state of the dogs remained stable pre- and post-session. Our result differs from other studies that have reported oxytocin level increases in dogs after positive interactions with humans (see, e.g., in [30,31,32,42,43]). Our methodology may explain why we did not observe an increase in oxytocin post-session: we did not want to interfere with the dynamics between the patients with FM and the AAA teams, and thus did not give any instructions to the patients on how to interact with the dogs. We believe that this better represents what therapy dogs are experiencing while participating in AAA or AAT sessions. However, this is different from the methodologies of most of the oxytocin literature cited, where the experiments are designed to maximize the time the human is in physical contact with the dog (e.g., sit on the floor and call the dog) and the way in which the human interacts with the dog is somewhat standardized (e.g., pet the dog in a specific way, make eye contact, and use a specific tone of voice).

The brain processes emotions asymmetrically [4,44,45]. Activity changes in the right hemisphere of the brain (indirectly measured by temperature changes in the ipsilateral tympanic membrane) offer important clues for understanding animal wellbeing as this hemisphere is involved in stress response [45]. Despite hemispheric laterality having been documented in many species [46], we did not find other studies where changes in tympanic membrane temperatures were used to measure stress response or wellbeing in dogs. Work points to the existence of hemispheric laterality in dogs [47]; similarly, in our study, the temperature of the right tympanic membrane of the dogs was significantly lower post-session. This suggests that the dogs were in a more relaxed state post-session compared with pre-session. Many factors may influence body temperature (e.g., age, sex, body size, ambient temperature, general health, and physical exertion). However, because the dogs in our study were well-trained, healthy adult canines that went through a standardized procedure in a temperature-controlled room, we believe that the changes in tympanic membrane temperature observed were caused by the experimental condition.

The autonomic nervous system (ANS) controls many cardiovascular parameters, such HR and HRV [48]. Under arousal (either positive or negative) the ANS becomes activated, which can lead to changes in HR and HRV [49]. Increased HRV is thought to be reflective of increased physical and psychological wellbeing [50], and because of this, HRV has been used as an indicator of wellbeing [48,49,51,52,53]. In a study on the effect of volunteers petting shelter dogs for 15 min, McGowan et al. [27] observed a decrease in HR and an increase in several components of HRV, including HF, PNN50, and RMSSD values, suggesting that the interactions had positive effects on the dogs. Researchers have identified that a decrease in SDNN (the standard deviation of normal RR intervals) can be associated with a decrease in sympathetic activity [54] while Katayama et al. [52] noted a decrease in SDNN during positive interactions between dogs and their owners. In our study, the dogs’ HR was significantly lower at the end of the session. Although not statistically significant, the HRV parameters HF and RMSSD were higher at the end of the session. Taken together, the observed changes in cardiovascular parameters suggest that the therapy dogs were in a calmer state after the therapy session.

## 5. Conclusions

Our results suggest that well-trained, registered therapy dogs are not negatively affected by their participation in AAA with patients with FM. Moreover, the dogs in our study were seemingly in a more relaxed/calm state following the AAA session. Most of the physiological parameters investigated showed no changes from pre- to post-session, and those that did change were in a direction indicative of a more positive emotional state by the end of the session.

The use of multiple noninvasive physiological indicators provides ways for researchers to accurately study and objectively measure the wellbeing of dogs involved in AAA. This helps to avoid the pitfalls associated with relying on too few physiological measures. It also allows researchers a chance to monitor therapy animals in clinical situations where the videotaping of the interaction between patients and the animals may not be ethically appropriate.

Our work adds to the existing body of research on the wellbeing of therapy dogs and suggests that AAA has little negative effects on well-trained, therapy dogs. However, wellbeing is more than the absence of negative effects. Positive wellbeing relies on identifying conditions in which animals thrive. We believe that additional research using multiple noninvasive physiological indicators should be conducted to identify which aspects therapy dogs find to be enjoyable during AAA. These findings could be used to develop evidence-based AAA standards to ensure the wellbeing of therapy animals.

## Figures and Tables

**Table 1 animals-10-00819-t001:** Description of the dogs involved in the study.

Dog	SEX	WT ^1^	AGE (yr)	BREED	AAT ^2^ Experience (yr)	CERTIFIED WITH
A	F	74	5	Golden	5	Helping Paws
B	F	65	3	Lab	0.3	Pet Partners
C	M	22	7	Chug	0.4	Pet Partners
Da	M	70	4	Wirehair Griffon	4.3	Alliance of Therapy Dogs
E	F	65	11	Golden Mix	8	TDI ^3^
F	F	42	5	Australian Shepherd	2	Pet Partners
Ga	F	45	4	Australian Shepherd	1	Pet Partners
H	M	60	2	Lab Mix	2.3	Alliance of Therapy Dogs
I	F	68	3	Golden	0.6	Pet Partners
J	M	26	10	Cockachon	0.25	Pet Partners
K	M	70	1.5	English Cream Golden	0.1	Pet Partners
L	F	25	6	Cocker Spaniel	1.2	Alliance of Therapy Dogs
M	M	92	4	Lab	2	Pet Partners
N	F	65	7	Standard Poodle	0.2	TDI ^3^
O	F	50	5	Golden	1.5	Pet Partners
Pa	F	60	4	Goldendoodle	0.3	Pet Partners
R	M	25	3	Mixed	0.3	Alliance of Therapy Dogs
Sa	M	70	4	Wirehair Griffon	0.3	Alliance of Therapy Dogs
T	F	60	12	Golden	10	TDI

^1^ Weight in pounds. ^2^ Animal-assisted therapy. ^3^ Therapy Dogs International.

**Table 2 animals-10-00819-t002:** Outline of general methods during a 20-min interaction with a therapy dog.

Event	Description
Pre-session (preparation room)	Saliva samples collected, using VersiSALs (Oasis, Vancouver, WA, USA) collection devices, for cortisol and oxytocin concentrations. Collections were within a 2-min timeframe.
	Tympanic membrane temperatures were collected, left and right ears simultaneously.
	The dog was fitted with the cardiac monitor.
Interaction—20 min (observation room)	The volunteer and dog entered the observation room where the patient was waiting.
	Dog was let off leash to move freely around the room.
	Dog and patient could choose to interact with one another.
	The volunteer could choose to sit in a chair or on the floor, according to their preference.
	A study staff member sat at a desk in the room, and did not interact with the patient unless to adjust the cardiac monitor.
	Continuous heart rate and heart rate variability were recorded throughout the entire interaction.
Post-session (preparation room)	The volunteer and the dog exited the observation room and went back into the preparation room.
	A saliva sample was collected, tympanic membrane temperatures were taken, and the cardiac monitor was removed.
	At the end, the dog and volunteer were free to go home.

**Table 3 animals-10-00819-t003:** Physiological parameters of therapy dogs pre- and post-20-min interaction with a fibromyalgia patient.

Parameters	Pre-Session						Post-Session						
	Dogs	n ^6^	Means	Min	Max	SD ^7^	Dogs	n ^6^	Means	Min	Max	SD ^7^	*p*-Value
Salivary Cortisol (µg/dL)	17	43	0.53	0.13	1.53	0.38	15	40	0.53	0.16	1.33	0.30	0.78
Salivary Oxytocin (nmol/L)	18	76	1.00	0.09	2.35	0.50	8	76	1.00	0.26	2.41	0.48	0.85
Right Tympanic Membrane Temperature (°C)	19	107	38.32	37.20	39.30	0.45	19	107	38.21	37.20	39.20	0.41	0.0036 ^a^
Left Tympanic Membrane Temperature (°C)	19	110	38.19	36.80	39.40	0.57	19	109	38.17	37.10	39.30	0.50	0.22
Heart Rate (bpm ^1^)	17	40	102.15	59.99	161.31	21.42	15	34	90.45	60.20	127.67	15.56	0.0003 ^a^
HF ^2^	17	40	954.23	0.05	2773.90	757.29	15	32	1090.49	6.30	4162.40	1057.09	0.56
RMSSD ^3^	17	42	54.80	0.00	110.43	24.11	15	33	60.17	8.68	121.05	32.46	0.13
PNN50 ^4^	17	44	27.63	0.00	73.11	16.84	15	36	29.06	0.00	66.66	20.88	0.51
LF/HF Ratio ^5^	17	42	2.12	0.06	6.92	1.88	15	33	1.89	0.20	6.74	1.50	0.63

^1^ Beats per minute. ^2^ High frequency. ^3^ The mean number of times per hour in which the change in consecutive normal sinus intervals exceeds 50 milliseconds. ^4^ The root square mean of the successive differences of RR intervals. ^5^ Low-high frequency power ratio. ^6^ Number of replicates. ^7^ Standard deviation. ^a^ Significantly different (*p* < 0.05).

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
