# Peer review of "Physiological State of Therapy Dogs during Animal-Assisted Activities in an Outpatient Setting"

_animals, 2020, doi:10.3390/ani10050819_

Round 1

Reviewer 1 Report

I want to start off with saying this is an excellent study. I have reviewed many studies that include animals and have never seen the quality and design so tight and well executed. One of the major flaws is the lack of information about the dogs and their qualifications along with their handler (if they had one). Your selection of dogs is excellent. As I worked with Pet Partners for several years and watched many, many dogs and their handlers go through vigorous testing, I feel you selected the right animals.

Your introduction is excellent, one of the finest I have read in dog studies. The effect, positive or negative, in the animals is such an import and often overlooked part of AAA/AAT, making your study innovative.

Your design is excellent and I was glad to see a description of the intervention, which is often lacking.

Excellent lit review.

One difficulty I had. Your study was a part of a larger study which had a treatment and control group, while yours was pre and posttest. On line 230 you mention the control group. I had to re-read and go back to other sections as I thought I missed something only to realize you were referring to the control group from the larger study. The way the sentence was there made it sound as if your study had a control group.  Perhaps you can reword this?  (as an aside, I have been house bound due to the virus for 30 days now and might not be so sharp!! at the moment)   

I had wondered if the dogs measures were affected by being in the new environment but then re-read that they were already volunteers at the facility.

I do believe in the conclusion was concise and succinct. I feel the first line of it was vital: well trained, registered therapy dogs. Wish all facilities utilizing animal would use these type of animals.

Bravo, well done. I am going to put accept with minor revision as perhaps you could clarify the sentence on line 230. Otherwise it is perfect in my opinion

Author Response

Reviewer #1

Comment #1: On line 230, you mention the control group. I had to re-read and go back to the other sections as I thought I missed something only to realize you were referring to the control group from the larger study. 

Thank you for mentioning this. We decided to remove the two sentences below since this methodology does not pertain to the current study, but, as you pointed out, to the larger study.

Research suggests that changes in emotional states may have an impact on oxytocin levels [20-22]. For this reason, for the control group, the volunteers were asked not to bring up conversations about pets or dogs in general, a potentially emotional topic for the patients with FM.

Reviewer 2 Report

To determine the effects on therapy dogs of participating in Animal-Assisted Activities (AAA), the authors recorded several physiological measures (e.g., salivary cortisol and oxytocin, tympanic membrane temperatures, heart rate and heart rate variability) both before and after 19 dogs participated in AAA sessions. They found no negatives effects of AAA on the dogs; heart rate and right tympanic membrane temperature were lower post session, suggesting the dogs were more relaxed after taking part in AAA. This study was part of a larger study assessing the impact of AAA on patients with fibromyalgia. The research question is interesting, and the results have application for assessing the welfare and wellbeing of therapy dogs. Strengths of the study include a nice sample size of dogs and use of several physiological measures. A weakness of the study is that the authors did not score the behavior of dogs during the AAA session (a study staff member was present in the room but only checked placement of the heart monitor). Scoring the behavior of dogs during the AAA session would have provided information on the extent and nature of interactions between dogs and participants, which could be factored into the analyses. My specific comments are detailed below.

The Introduction is very long and should be shortened. The studies described from lines 95-172 should be summarized rather than providing the current level of detail for each one (e.g., each study currently has its own paragraph).

Lines 181-187: In this paragraph, explicitly state how your study differs from previous studies (i.e., what is new?).

Lines 211 and 214: Is the owner always the volunteer? Please clarify.

Lines 249-251: Again, the study would have been more complete if the staff member present in the room recorded behavior of the dogs during the AAA.

Results section: There is no need to repeat the means and SDs for all of the physiological measures in the text (these values are all in Table 2); just report the values for F, df, and p in the text.

Discussion: How did your physiological measures (e.g., cortisol and oxytocin concentrations, heart rate variability) generally compare to those already in the literature? Were your values similar to or different from those previously reported? If similar, this would assure readers that the methods used resulted in measurements comparable to those in the literature.

Minor points:
Line 16: Change “on” to “to” and “has” to “have” (therapy dogs and their benefits is plural)
Line 72: Change “have” to “has” (work is singular)
Line 79: Citation needs editing.
Line 240: Change “chose” to “choose”
Table 1, page 6: The phrase “interact with the interaction” is awkward. Also, change “was” to “were” (continuous heart rate and heart rate variability are plural)
Lines 295-297: Data is plural, so change “data was” to “data were” in these sentences.
Line 360 Requires editing (“we posed to answer this question”)
Line 418: Change “during the session” to “from pre- to post-session” to be more accurate

Author Response

Reviewer #2

Comment #1: A weakness of the study is that the authors did not score the behavior of dogs during the AAA session (a study staff member was present in the room but only checked placement of the heart monitor). Scoring the behavior of dogs during the AAA session would have provided information on the extent and nature of interactions between dogs and participants, which could be factored into the analyses.

We realize that this is a limitation of our study. We added the following language this the discussion section (now at line 364).

One limitation of this study is the fact that the dogs’ behavior was not recorded or scored during the sessions. Behavioral observations would have provided additional information on the extent and nature of the interactions between the dogs and the patients with FM. This information would have helped better interpret the physiological data.

Comment #2: The Introduction is very long and should be shortened. The studies described from lines 95-172 should be summarized rather than providing the current level of detail for each one (e.g., each study currently has its own paragraph).

Because there is so much variation between studies in terms of study designs, participants, conditions studied, etc., we believe that providing this level of information gives the readers the ability to better understand the findings of the studies and helps put them in context. Another reviewer for this paper mentioned that our “introduction is excellent, one of the finest I have read in dog studies. The effect, positive or negative, in the animals is such an import and often overlooked part of AAA/AAT, making your study innovative. Excellent lit review.” However, if the editor would like us to shorten this section, we would be happy to comply.

Comment #3: Lines 181-187: In this paragraph, explicitly state how your study differs from previous studies (i.e., what is new?).

We added the following sentence (line 184).

To our knowledge, this combination of methodologies (including a new salivary oxytocin analysis developed for this project) has not been used before.

Comment #4: Lines 211 and 214: Is the owner always the volunteer? Please clarify.

We added the following language to clarify.

The dogs were always accompanied by their owners (referred to as “volunteers” later in the text) (line 210)

 Comment #5: Results section: There is no need to repeat the means and SDs for all of the physiological measures in the text (these values are all in Table 2); just report the values for F, df, and p in the text.

We removed means and SDs in the text of the Results section.

Comment #6: Discussion: How did your physiological measures (e.g., cortisol and oxytocin concentrations, heart rate variability) generally compare to those already in the literature? Were your values similar to or different from those previously reported? If similar, this would assure readers that the methods used resulted in measurements comparable to those in the literature.

We added the following paragraph in the discussion section (line 368).

The mean cortisol concentration levels pre-session for the dogs in our study were comparable to concentration levels reported in the literature [see review 34]. Less is known about oxytocin in dogs. The available data show greater variation [39,40,43]. This may be to the fact that individual dogs’ oxytocin levels have a wide range.  In addition, researchers are using different detection and quantification methodologies, and it is known that some methods are more sensitive than others [25]. Overall, our cortisol and oxytocin results suggest that pre-experiment variables, such as anticipation or transportation, did not affect the concentration of these hormones in the dogs.

Comment #7: Line 16: Change “on” to “to” and “has” to “have” (therapy dogs and their benefits is plural)

Changes made.

Therapy dogs and their benefits to human health have been studied extensively.

Comment #8: Line 72: Change “have” to “has” (work is singular)

Change made.

Work by animal welfare and behavior scientists in the field of AAA over the last decade or so has helped objectively measure the wellbeing of these animals.

Comment #9: Line 79: Citation needs editing.

We changed the sentence to:

Furthermore, inadequate environmental conditions, high temperatures, lack of space, and age of clients have been reported as potential factors negatively influencing the wellbeing of therapy dogs [9].

Comment #10: Line 240: Change “chose” to “choose”

(…)  the participants could choose to interact or not (…)

Comment #11: Table 1, page 6: The phrase “interact with the interaction” is awkward. Also, change “was” to “were” (continuous heart rate and heart rate variability are plural)

We changed the sentence to:

A study staff member sat at a desk, in the room, and did not interact with the participants unless to fix the cardiac monitor.

We also changed “was” for “were.”

Continuous heart rate and heart rate variability were recorded throughout the entire interaction.

Comment #12: Lines 295-297: Data is plural, so change “data was” to “data were” in these sentences.

We made the changes throughout the paragraph. (now starts at line 298)

The data were downloaded from the receiver to a computer using the Polar Flow application. The data were then exported from the Polar Flow (Polar Electro Öy, Kempele, Finland) application for analysis. The collected data were analyzed in two-minute intervals via Kubios HRV Standard Version 3.1.0 (Kubios Oy). The cardiac parameters were analyzed at the beginning of the session (minute 3 and 4) and the end of the session (minute 17 and 18); these timeframes were selected to provide the cleanest two-minute intervals for analysis, as this gave the dog time to settle at the beginning of the session and standardized a time point before the end of the session. Cardiac data that had an artifact correction factor of greater than 10% were not included in the analysis; this was done to minimize corruption in the data, accounting for motion artifacts and interference artifacts, such as the dog’s fur.

Comment 13: Line 360 Requires editing (“we posed to answer this question”)

We changed the wording of the sentence.

We investigated this question by evaluating (…)

Comment 14: Line 418: Change “during the session” to “from pre- to post-session” to be more accurate

Change made.

(…) showed no changes from pre- to post-session session (…)

Reviewer 3 Report

Interesting paper with a promising multifaceted approach. I only have a few comments on the method part (also connected to the discussion and conclusions) concerning oxytocin and cortisol.

e.g. 218 Exactly how long time after the session was the post-saliva sample obtained? If directly after the session, the post-oxytocin/cortisol results only reflect the very beginning of the session (since it takes 15-20 min for oxytocin/cortisol to increase in the saliva) and then the authors need to adjust their discussion and conclusion that these results reflect 20min session (which they then do not).

e.g. 236 Here the authors write patients, but later they write participants. Try to be consistent.

Also, in the discussion part - the authors need to compare their oxytocin/cortisol levels to other studies to show that their levels were not increased already before the session (due to e.g. transport associated with the session).

Author Response

Reviewer #3

Comment #1: e.g. 218 Exactly how long time after the session was the post-saliva sample obtained? If directly after the session, the post-oxytocin/cortisol results only reflect the very beginning of the session (since it takes 15-20 min for oxytocin/cortisol to increase in the saliva) and then the authors need to adjust their discussion and conclusion that these results reflect 20min session (which they then do not).

There is some controversy in the literature over the ideal timing for saliva sampling for analysis of cortisol. Most notably, Vincent and Michell (1992) reported evidence of a delay in the increase of cortisol in saliva compared with blood and, by contrast, Beerda et al. (1996) detected no delay in salivary cortisol responses with respect to plasma responses. In addition, Beerda et al. (1998) succesfully measured salivary cortisol changes 10, 15, 20, and 30 minutes after dogs were submitted to an experimental condition.  It has been suggested (Kobelt et al., 2003) that these differences may arise depending on whether whole serum or plasma concentrations are compared with saliva, as saliva only contains the free fraction of cortisol. The transfer of cortisol from plasma to saliva occurs very rapidly. Within less than a minute, cortisol injected intravenously appears in saliva, and the peak concentration in saliva lags by less than 2–3 min compared to levels measured in the blood (Kirschbaum and Hellhammer, 2000). The advantages of salivary cortisol sampling have been summarized in several reviews (e.g., Kirschbaum and Hellhammer, 1994). In essence the method is preferred by many as it allows for relatively non-invasive sample collection, and salivary cortisol represents the biologically active fraction of the hormone (Vinning et al., 1983; Mendel, 1989). Salivary cortisol is ideal for assessing acute responses to stimuli (Nicolson, 2007). However, salivary cortisol returns to baseline more slowly than blood after psychosocial stressors (Kirschbaum and Hellhammer, 2000) so the ideal timing of samples aimed to detect a decline in cortisol rather than heighted cortisol response after a stressor are not well known. This being said, it is possible that a longer delay in sampling time after the AAA session may have resulted is slightly different salivary cortisol concentrations. More work in this area is necessary to determine the most appropriate timing for collection of saliva following a positive event (rather than a stressor) for the best assessment of the deactivation (as opposed to activation) of the hypothalamic-pituitary-adrenal axis. Having said that, measuring  salivary cortisol after a short (20 min or less) period is commonly done in humans and dogs (see for example, Krause-Parello et al 2012; Naumova et al. 2015; Ooishi et al. 2017).

Comment #2: e.g. 236 Here the authors write patients, but later they write participants. Try to be consistent.

We changed most of the word “participant” to “patient.” However, on 3 occasions it seemed more appropriate to refer to “participants.”

Comment #3: Also, in the discussion part - the authors need to compare their oxytocin/cortisol levels to other studies to show that their levels were not increased already before the session (due to e.g. transport associated with the session).

We added the following paragraph in the discussion section (line 368).

The mean cortisol concentration levels pre-session for the dogs in our study were comparable to concentration levels reported in the literature [see review 34]. Less is known about oxytocin in dogs. The available data show greater variation [39,40,43]. This may be to the fact that individual dogs’ oxytocin levels have a wide range.  In addition, researchers are using different detection and quantification methodologies, and it is known that some methods are more sensitive than others [25]. Overall, our cortisol and oxytocin results suggest that pre-experiment variables, such as anticipation or transportation, did not affect the concentration of these hormones in the dogs.

Reviewer 4 Report

This study was well designed and conducted. They had a fair sample size with pre and post measures. One very good aspect of this study was that the control group utilized the same volunteers without dogs.

The title may be misleading – this was more about the physiologic state of therapy dogs than the specific emotional state. While these physiologic factors taken together may suggest an emotional state, I don’t know that it can be said definitively. If the authors decide to keep this title, the introduction and conclusion should further elaborate and be explicit in how each of these factors translate to emotion and the reasoning behind it.

Limitation: The study is lacking the information about what happened during the actual interaction, which is likely a key factor in how these dogs responded. There appears to be a missed opportunity in the study: If a study observer was present for every interaction, this individual could have also reported some description of the level of interaction, which would have added significantly to the study. Additionally, behavior could have been observed before and after sessions.

211 – What exactly is the breed; include table of age, weight, breed, sex, how long the dog had been registered and how long volunteering at that location? This would be helpful in table format.  

214 – How long after arrival did the dog sit before samples taken? Travel and initial excitement and acclimating to the environment may impact these values.

246 – Size or room. What was the average ambient temperature (which may impact tympanic temperature results)?

251 – how often did the study staff member need to intervene?

378 – Are these oxytocin values comparable to other studies in dogs?

391-399 – What other factors can impact temperature other than stress? Disease, ambient temperature, excitement, etc.

There is evidence to suggest that cortisol values are lower when the dogs are off-leash.

Glenk LM, Kothgassner OD, Stetina BU, Palme R, Kepplinger B, Baran H. Therapy dogs' salivary cortisol levels vary during animal-assisted interventions. Animal Welfare. 2013;22(3):369-78.

Please elaborate on the implications of these findings, as  most AAI organizations require dogs to be on leash at all times.

Limitations of the study? A significant one would be not including behavioral observations. Although there was explanation of not observing behavior during the actual session, behavioral observation could have been recorded before and after.

Author Response

Reviewer #4

Comment #1: The title may be misleading – this was more about the physiologic state of therapy dogs than the specific emotional state. While these physiologic factors taken together may suggest an emotional state, I don’t know that it can be said definitively. If the authors decide to keep this title, the introduction and conclusion should further elaborate and be explicit in how each of these factors translate to emotion and the reasoning behind it.

We changed the word “emotional” by “physiological” in the title.

We also made the same change the sentence on line 355, for consistency.

The present study aimed to answer the following question: Is the physiological state of well-trained therapy dogs impacted by a 20-minute animal-assisted activity?

 Comment #2: Limitation: The study is lacking the information about what happened during the actual interaction, which is likely a key factor in how these dogs responded. There appears to be a missed opportunity in the study: If a study observer was present for every interaction, this individual could have also reported some description of the level of interaction, which would have added significantly to the study. Additionally, behavior could have been observed before and after sessions.

We realize that this is a limitation of our study. We added the following language this the discussion section (now at line 360).

One limitation of this study is the fact that the dogs’ behavior was not recorded or scored during the sessions. Behavioral observations would have provided additional information on the extent and nature of the interactions between the dogs and the patients with FM. This information would have helped better interpret the physiological data.

Comment #3: 211 – What exactly is the breed; include table of age, weight, breed, sex, how long the dog had been registered and how long volunteering at that location? This would be helpful in table format. 

We added a table that provides information on the dogs in this study. Inserted on line 216 (now Table 1). We renumbered the other tables accordingly.

We removed the sentence below, as it was not necessary anymore.

The utilized breeds included Golden Retrievers, Australian Shepherd, Wirehair Griffon, Labrador Retriever, Standard Poodle, Cocker Spaniel, and mixed breeds, such as Goldendoodle, Chug, and Cock-a-chon.

Table 1. Description of the dogs involved in the study

Dog ID

SEX

WT

AGE

BREED

AAT experience (yr)

CERTIFIED WITH

A

F

74

5

Golden

5

Helping Paws

B

F

65

3

Lab

0.3

Pet Partners

C

M

22

7

Chug

0.4

Pet Partners

Da

M

70

4

Wirehair Griffon

4.3

Alliance of Therapy Dogs

E

F

65

11

Golden Mix

8

TDI

F

F

42

5

Australian Shepherd

2

Pet Partners

Ga

F

45

4

Australian Shepherd

1

Pet Partners

H

M

60

2

Lab Mix

2.3

Alliance of Therapy Dogs

I

F

68

3

Golden

0.6

Pet Partners

J

M

26

10

Cockachon

0.25

Pet Partners

K

M

70

1.5

English Cream Golden

0.1

Pet Partners

L

F

25

6

Cocker Spaniel 

1.2

Alliance of Therapy Dogs

M

M

92

4

Lab

2

Pet Partners

N

F

65

7

Standard Poodle

0.2

TDI

O

F

50

5

Golden

1.5

Pet Partnerts

Pa

F

60

4

Goldendoodle

0.3

Pet Partnerts

R

M

25

3

Mixed

0.3

Alliance of Therapy Dogs

Sa

M

70

4

Wirehair Griffon

0.3

Alliance of Therapy Dogs

T

F

60

12

Golden

10

TDI

Comment #4: 214 – How long after arrival did the dog sit before samples taken? Travel and initial excitement and acclimating to the environment may impact these values.

We added this information and the sentence now reads (line 216):

About 15 minutes after arriving at the hospital and before meeting the patients with FM in the exam room, the dog and volunteer were brought to an adjoining room where the dog’s saliva was collected, bilateral tympanic membrane temperatures were taken simultaneously, and a heart rate monitor was placed on the dog by a study team member.

 Comment #5: 246 – Size or room. What was the average ambient temperature (which may impact tympanic temperature results)?

We added this information to the sentence (line 247). It now reads:

The study interactions took place in an exam room (8’x 12’; average room temperature: 69°F) in Mayo Clinic’s Fibromyalgia and Chronic Fatigue Clinic, which included a small exam table, four chairs, and a desk.

 Comment #6: 251 – how often did the study staff member need to intervene?

We added this information and the sentence now reads (line 250):

The study staff member sat at a desk in the room to monitor the interaction and was only allowed to interact with the dog if the heart rate monitor was not functioning properly (the staff member intervened 4 times to address lost signal on the heart rate monitor, and once to stop a conversation that fell outside of the approved topics).

 Comment #7: 378 – Are these oxytocin values comparable to other studies in dogs?

We added the following paragraph in the discussion section (line 368).

The mean cortisol concentration levels pre-session for the dogs in our study were comparable to concentration levels reported in the literature [see review 34]. Less is known about oxytocin in dogs. The available data show greater variation [39,40,43]. This may be to the fact that individual dogs’ oxytocin levels have a wide range.  In addition, researchers are using different detection and quantification methodologies, and it is known that some methods are more sensitive than others [25]. Overall, our cortisol and oxytocin results suggest that pre-experiment variables, such as anticipation or transportation, did not affect the concentration of these hormones in the dogs.

Comment #8: 391-399 – What other factors can impact temperature other than stress? Disease, ambient temperature, excitement, etc.

We added the following sentence on line 406.

Many factors may influence body temperature (e.g., age, sex, body size, ambient temperature, general health, physical exertion). However, because the dogs in our study were well-trained, healthy adult canines that went through a standardized procedure in a temperature-controlled room, we believe that the changes in tympanic membrane temperature observed were caused by the experimental condition.

Comment #9: There is evidence to suggest that cortisol values are lower when the dogs are off-leash.

Glenk LM, Kothgassner OD, Stetina BU, Palme R, Kepplinger B, Baran H. Therapy dogs' salivary cortisol levels vary during animal-assisted interventions. Animal Welfare. 2013;22(3):369-78.

Please elaborate on the implications of these findings, as most AAI organizations require dogs to be on leash at all times.

Conditions in which AAA sessions are performed (including requiring dogs to be on leash at all time) certainly can impact the wellbeing of the therapy dogs and are worth investigating. However, since we did not compare the physiological responses of therapy dogs when they were off leash versus when they were on leash, we do not believe that it would be appropriate for us to discuss this recommendation.